# Pesticides and transgenerational inheritance of pathologies: Designing, analysing and reporting rodent studies

Ian Plewis *

Social Statistics, School of Social Sciences, University of Manchester, Manchester, United Kingdom

* ian.plewis@manchester.ac.uk

**Data Availability Statement:** All relevant data are within the manuscript and its Supporting Information files.

**Funding:** The author(s) received no specific funding for this work.

## Abstract

Single-centre studies examining the transgenerational inheritance of pathologies in rodents exposed to pesticides have not always taken important design and analysis issues into account. This paper examines these methodological and statistical issues in detail. Its particular focus is on the estimation of 'litter effects': the tendency for rodents within a litter to be more alike than rodents in different litters. Appropriate statistical models were fitted to published data from a series of widely reported studies carried out at Washington State University. These studies were amalgamated into a single dataset in order to estimate these litter effects and associated treatment effects. Litter effects varied by outcome and were often substantial. Consequently, the effective sample size was often substantially less than the number of observations with implications for the power of the studies. Moreover, the reported precision of the estimates of treatment effects was too low. These problems are exacerbated by unexplained missing data across generations. Researchers in the life sciences could be more cognisant of the guidelines established in medicine for reporting randomised controlled trials, particularly cluster randomised trials. More attention should be paid to the design and analysis of multi-generational rodent studies; their imperfections have important implications for assessments of the evidence relating to the risks of pesticides for public health.

## Introduction

The reproductive toxicity and carcinogenicity of pesticides are widely assessed via feeding experiments using rodents. This paper is a critical examination of the methodological and statistical aspects of multi-and transgenerational studies investigating adverse effects in pesticide-exposed rodents. In some studies, all generations are exposed but to be classified as a transgenerational study just the first generation is exposed with a focus on any adverse effects in subsequent unexposed generations [1]. An important, and widely ignored, issue in the analysis of these studies is the lack of independence between observational units after the first generation. This is often referred to as the 'litter effect': the tendency of two randomly selected rodents from the same litter to be more alike than two randomly selected rodents from different litters. Ignoring litter effects can lead to under-powered experiments and biased statistical inferences

**Competing interests:** The authors have declared that no competing interests exist.

as we see from a reanalysis of data from five papers from the Center for Reproductive Biology, Washington State University (WSU), USA [2–6]. Each of these WSU papers analyses, in essentially the same way, epigenetic transgenerational inheritance after rats have been exposed to pesticides: atrazine and glyphosate (herbicides), methoxychlor and a mixture of permethrin and DEET (insecticides) and vinclozolin (fungicide). Uncommonly, the authors include some raw data in the supporting information that accompanies their papers. Some of the design and analysis issues for studies of this kind can therefore be drawn out, and empirical evidence about the size of litter effects can be provided.

Statisticians are familiar with the problems generated by observational dependence, and the opportunities provided by it [7], but these statistical techniques are not widely used in the life sciences literatures. To give two examples from toxicology, both studying the effects of the herbicide glyphosate: the first [8] analysed data generated by an experiment with 108 F1 offspring from just 24 F0 dams but ignored potential litter effects at F1; the second [9] used a multi-generational design where the sample sizes are 21 for F0, 65 for F1 and 482 for F2 but made no allowance for either the clustering of F1 animals within the F0 dams or the clustering of F2 offspring within both grandparents (F0) and parents (F1). A rare counterexample [10] allows for litter effects in the analysis of a two-generational study but does not estimate them. In addition:

a.  The 'litter effect' is correctly identified in [1, 11, 12]. However, it is then argued that when treatments are either directly or indirectly assigned to litters, the experimental unit should be the litter rather than the individual animal. In fact, as we shall see, a more appropriate analysis is one that focuses on the individual animals–each of which will respond differently to the treatment—but which takes account of the dependence within litters. The same misconception emerges in a different context–the housing of two or more rodents within a cage–generating a cage effect where it is mistakenly argued that if treatments are allocated to cages then the cage, and not the individual animal, should be the experimental unit [13].

b.  In a wide-ranging critique of experimental methods in neuroscience [14], the litter effect is also identified as an issue and the authors correctly recognise that so-called mixed models are the most appropriate method of analysis. Their advice on how to estimate treatment effects when the treatment is allocated to a litter is, however, rather ambiguous. A similar paper [15] describes how litter effects are ignored in behavioural neuroendocrinology. These two papers focus on the implications of ignoring litter effects on statistical inference, both in terms of generating false positive treatment effects (i.e. incorrect Type 1 errors) and reducing power (increasing Type 2 errors).

The overall aim of the paper is to alert researchers to the dangers of ignoring litter effects by demonstrating how substantial they can be. It extends the literature in that litter effects are estimated using appropriate statistical models and their influence on treatment effects demonstrated. The analyses cast doubt on the validity of some of the conclusions from the WSU studies but they also have implications for the conduct of rodent studies of all kinds as well as for policy decisions about the risks attached to pesticides.

## Designing transgenerational studies

Transgenerational rodent studies face a number of design issues, discussed in detail in [16]. They follow the design protocol set out below, or variants of it:

i.  Rodents, usually pregnant females (F0), are assigned, ideally randomly, to one or more treatment groups and a control, thereby generating two or more experimental groups. A

fundamental aspect of any transgenerational design is that the treatment is administered (or not) only to the F0 females and later generations belong to the experimental groups solely as a result of this initial administration. This has implications for the power of these studies as discussed below.

ii. Various measures of some or all their offspring (F1) are taken.

iii. To generate F2 offspring, F1 rodents from all the experimental groups are either mated with rodents from different litters but from the same F0 experimental group ('dual breeding') [2–6], or exposed females are mated with unexposed males from outside the study ('matriline') [9], or exposed males are mated with unexposed females from outside the study ('patriline') [17]. All or some of these offspring are measured. The implications for statistical analysis of these different mating arrangements do not appear to have been discussed in the literature.

iv. Stage (iii) is repeated for all the subsequent generations in the design.

The validity of any randomisation at F0 is compromised if an F0 female is not represented in all subsequent generations. These problems arising from self-selection after F1 can be magnified by the ways in which rodents within litters are selected for study. If this is done randomly in such a way that the same proportion of rodents from each litter are measured then this is satisfactory. Often, however, it is difficult to know from published reports of studies just why some rodents are measured and others not, and what the effects of these different sources of missing data might be.

Turning to the design of the WSU studies [2–6], a careful reading of them indicates that they did not use random assignment; there is an imbalance between the numbers of treatment and control rats at F1 and some controls appear in the same three studies (S1 Table, note 5). Failure to randomise does, of course, cast doubt on any attribution of group differences to the treatment in question. This doubt is reinforced when we see (S1 Table) that the F0 females become progressively under-represented in all five studies as we move down the generations and this is particularly marked in the control groups. The overlap in membership between the control groups–and the fact that each of the five studies address essentially the same research question in the same way–means that, from the methodological perspective adopted here, it is both reasonable and efficient to combine the five studies into a single re-analysis with a set of outcomes defined in S1 Appendix.

## Analysing observational dependence

Litter effects arise when the offspring (Fx, x ≥ 1) of pregnant females (F0) are studied. Plausible statistical models for the analysis of continuous and binary outcomes at Fx, x ≥ 1 –known variously as multilevel, hierarchical, random effects and mixed models—are set out in S2 Appendix (models 1a and 1b). These analyses are only possible if sufficient identifying information is provided at each generation, a point I return to below.

Note that if treatments are assigned at random to rodents *within* a litter then litter becomes what is known as a blocking or control variable or factor that should be included in the analysis as it will be expected to increase power [11]. I do not deal with this situation here. There are a few studies [e.g. 17, 18] in which researchers aim to avoid litter effects by randomly selecting one rodent (or one male and one female, analysed separately) from each litter. This requires a much larger pool of rodents at F0 but does avoid the difficulties posed by observational dependence in adjacent generations although not necessarily in generations further apart. In the majority of experiments, however, some or all the rodents within a litter are measured and

**Table 1. Litter effects (95% CI) [sample sizes] by outcome.**

| OUTCOME | F1:F0[1] | F2:F0 | F2:F1 | F3:F0 | F3:F2 |
|---|---|---|---|---|---|
| Puberty | 0.67[2] (0.44–0.85) [368:61] | 0.61 (0.36–0.82) [301:28] | 0.73 (0.52–0.87) [301:49] | 0.62 (0.39–0.81) [667:45] | 0.81 (0.70–0.88) [667:121] |
| Testis | 0.40 (0.16–0.70) [167:53] | 0[3] [115:25] | 0 [115:39] | 0.12 (0.03–0.40) [306:44] | 0.24 (0.09–0.51) [306:114] |
| Ovary | 0.25 (0.06–0.63) [149:45] | 0 [123:23] | 0.31 (0.09–0.66) [123:36] | 0.34 (0.13–0.64) [192:34] | 0.33 (0.12–0.64) [192:78] |
| Prostate | 0 [170:52] | 0 [107:23] | 0 [107:37] | 0.11 (0.02–0.39) [302:44] | 0.25 (0.08–0.55) [302:115] |
| Kidney | 0.28 (0.13–0.50) [339:57] | 0.21 (0.06–0.52) [259:24] | 0.24 (0.06–0.60) [259:43] | 0.18 (0.08–0.36) [563:44] | 0.27 (0.14–0.46) [563:119] |
| Lean | 0.51 (0.28–0.73) [271:39] | 0.33 (0.13–0.61) [380:32] | 0.40 (0.19–0.66) [380:55] | 0.35 (0.17–0.59) [404:28] | 0.49 (0.31–0.68) [404:55] |
| Obese | 0.47 (0.29–0.67) [357:51] | 0.36 (0.19–0.57) [380:32] | 0.36 (0.20–0.56) [380:55] | 0.32 (0.19–0.50) [642:43] | 0.39 (0.25–0.56) '[642:111] |

Notes

[1] Column headings: Fx:Fy signifies generation x (x = 1,2,3) nested within generation y (y = 0,1,2).

[2] Each cell gives the estimated litter effect (i.e. the intra-litter correlation), its 95% confidence interval and the sample sizes for the two levels (rat: litter).

[3] Estimated litter effects of zero indicate that the hypothesis of no litter effect could not be rejected (p > 0.05).

used in the analysis and it is then that the observational dependence that is central to this paper is generated and must be allowed for.

All results in this section come from the longitudinal ('xt') and mixed effects ('me') procedures in STATA [19] applied to a combined dataset of all the data (treatment and control; males and females) from all five WSU studies. The estimated litter effects—defined as the intra-litter correlation and measuring the similarity of rats within a litter (S2 Appendix)—are given in Table 1. They are based on (i) the null hypothesis of no treatment effects which is reasonable for F1 and F2 but less so for F3 (see below); (ii) two-level models as defined by the column headings because identifying information is not provided to enable the more appropriate four-level models at F3 to be estimated and because there are too few F0 observations to separate the variation at F0 and F1 (for F2) or at F0 and F2 (for F3); (iii) males and females combined as the sex differences in the prevalence of problems are generally small.

We see from Table 1 that the litter effects are usually greater for offspring nested within the previous generation (i.e. Fx:Fx-1) compared with generations separated by at least one intervening generation (e.g. Fx:Fx-2). The estimated litter effects are substantial for those three outcomes–puberty (between 0.61 and 0.81 across the five columns), lean (0.33–0.51) and obese (0.32–0.47)–where one might expect a strong genetic component. They are smaller for kidney (0.18–0.28), and sometimes zero for problems in the other three organs. Note, however, that, despite combining the datasets, the confidence intervals for the intra-litter correlations in Table 1 are often wide and show that precise estimation of litter effects for pathologies will often only be possible with large samples.

The stronger the litter effects the smaller the number of independent observations as shown in Table 2 for two of the outcomes–puberty and obese–where the litter effects and sample sizes are largest. The number of observations (n) from the reported data is contrasted with the effective sample size (ESS) at F3 where:

$$ESS = n/(1 + (m - 1)\hat{\rho})$$

$m$ is the average litter size and $\hat{\rho}$ is the estimated intra-litter correlation.

Table 2 shows just how much smaller is the range of the effective sample size compared with the number of observations so, for example, ESS ranges from 14 to 18 for puberty and atrazine compared with an observed sample size of 114. Hence, the power of the studies is reduced and this has implications for the estimation of any treatment effects. We also see that the total ESS is greater than the number of litters (e.g. 134 to 160 compared with 121 litters for

**Table 2. Numbers of observations and effective sample sizes at F3 by group and outcome.**

| GROUP | PUBERTY | | OBESE | |
|---|---|---|---|---|
| | n | ESS[1] | n | ESS[1] |
| Atrazine | 114 | 14–18 | 114 | 21–38 |
| Glyphosate | 52 | 10–12 | 75 | 15–27 |
| Vinclozolin | 103 | 16–19 | 91 | 25–41 |
| Methoxychlor | 59 | 20–23 | 59 | 26–37 |
| Permethrin+DEET | 68 | 14–17 | n.a.[2] | n.a. |
| Control | 271 | 60–71 | 303 | 87–144 |
| Total | 615 | 134–160 | 642 | 174–287 |

Note

[1] The range of the effective sample size (ESS) is determined by the upper and lower points of the confidence intervals for ρ in Table 1.

[2] Obesity data omitted—see S1 Appendix.

puberty from the final column of Table 1), implying that an analysis based on litter as the experimental unit will generate less precise estimates than one that uses the data from all rodents.

We would expect the lower effective sample size to lead to less precisely estimated treatment effects. This is illustrated for treatment effects at F3 –the focus of the WSU papers–allowing for litter effects at F2, and focusing on puberty (Table 3) and obese (Table 4) as in Table 2. As explained in S2 Appendix, there are two ways of estimating treatment effects that take account of litter effects: population average and multilevel (i.e. random effects) models. In addition, it is possible (although not advisable) to use the litter as the unit of analysis. The outcome is then the proportion of problems in a litter and a single-level model is fitted with a parameter that allows for extra-binomial variation generated by the unmeasured litter variables. Although the estimates from these models are different, particularly for puberty where the litter effects (see Table 1) are greater, we find a consistent reduction in the precision of the estimated treatment effects (i.e. wider confidence intervals) when compared with single-level logit models that

**Table 3. Puberty: F3 treatment Effects (95% CI).**

| GROUP | LOGIT[1] | LOGIT (LITTER)[2] | LOGIT (PA)[3] | LOGIT (RE)[4] |
|---|---|---|---|---|
| Atrazine | 1.0 (0.38–1.7) | 1.2 (0.03–2.4) | 0.92 (-0.43–2.3) | 1.8 (-0.67–4.3) |
| Glyphosate | n.e.[5] | n.e. | n.e. | n.e. |
| Vinclozolin | -2.1 (-4.1 - -0.07) | -2.0 (-5.6–1.6) | 0[6] | 0 |
| Methoxychlor | 0 | 0 | 0 | 0 |
| Permethrin+DEET | 1.9 (1.2–2.5) | 2.0 (0.80–3.2) | 1.8 (0.59–3.0) | 3.9 (1.4–6.5) |
| Litter effect (95% CI) | n.a. | n.a. | n.a. | 0.77 (0.60–0.87) |
| Sample size | 615 | 112 | 615:112 | 615:112 |

Notes

[1] Logistic regression ignoring litter effects. Each cell shows the log odds (and 95% CI) of a problem compared with the controls.

[2] Logistic regression with proportion of problems per litter as the outcome and including a parameter for extra-binomial variation = 3.2 (0.43).

[3] Population average logistic regression allowing for F2 litter effects.

[4] Random effects logistic regression allowing for F2 litter effects.

[5] n.e.–not estimable from the model as no observed problems at F3.

[6] Treatment effect set to zero if z-statistic < 1.

**Table 4. Obese: F3 Treatment Effects (95% CI).**

| GROUP | LOGIT[(1)] | LOGIT (LITTER)[(2)] | LOGIT (PA)[(3)] | LOGIT (RE)[(4)] |
|---|---|---|---|---|
| Atrazine | -1.1 (-2.2 - -0.02) | -1.1 (-2.3–0.19) | -1.0 (-2.3–0.33) | -1.1 (-2.4–0.13) |
| Glyphosate | 1.8 (1.2–2.4) | 1.9 (1.2–2.6) | 1.8 (1.1–2.6) | 2.1 (1.2–3.1) |
| Vinclozolin | 1.2 (0.63–1.9) | 1.2 (0.44–1.9) | 1.2 (0.45–1.9) | 1.3 (0.47–2.2) |
| Methoxychlor | 1.6 (0.98–2.3) | 1.7 (0.88–2.4) | 1.6 (0.87–2.4) | 1.8 (0.93–2.7) |
| Permethrin+DEET | n.e.[(5)] | n.e. | n.e. | n.e. |
| Litter effect (95% CI) | n.a. | n.a. | n.a. | 0.19 (0.08–0.40) |
| Sample size | 642 | 111 | 642:111 | 642:111 |

Notes

See Table 3. The estimate of the extra-binomial parameter = 1.4 (0.19). Also, all models for obesity include sex as an explanatory variable as females are less likely to be obese according to the method used to determine obesity as described in [4].

make no allowance for the structure of the data. For example, the confidence interval for the effect of atrazine on both puberty and obesity includes zero, and the surprising negative estimate for vinclozolin for puberty is shown to be essentially zero, after allowing for litter effects. The intra-litter correlations are 5% smaller in Table 3 and 51% smaller in Table 4 than they are in Table 1; this shows what proportion of the litter effects can be explained by the F0 treatments (and sex for obesity) and also that they are still important after allowing for these variables.

Models 1(a) and 1(b) in S2 Appendix are strictly hierarchical models but these models will not accurately reflect the structure of the experimental design if a dual breeding design is used. An F2 rodent will then belong to not just one but two F0 grandmothers, one via the maternal line and one via the paternal line (a structure that becomes more complicated at Fx, x > 2). A multiple membership model [20] is then required but it will only be possible to estimate the parameters of such a model if the full pedigrees of all the experimental animals are provided. Simulations [21] indicate that higher-level variances will be under-estimated when a multiple membership structure is analysed as if it were strictly hierarchical which in turn implies that estimates of litter effects (i.e. intra-litter correlations) and the precision of estimates of treatment effects will be too low. This needs to be borne in mind when assessing the results in Tables 1–4.

## Reporting standards

The ARRIVE2.0 guidelines–released on 14 July 2020 (https://arriveguidelines.org)–list 10 essential and 11 recommended guidelines for reporting animal studies. An explanation and elaboration of these guidelines can be found in [22]. They are not dissimilar to those set out in the CONSORT2010 statement [23] for reporting randomised controlled trials in medicine and health care. As the authors of the guidelines make clear, the conduct and reporting of studies using live animals often fall short of good practice. A particular reporting concern is the failure to account for missing data ('inclusion and exclusion criteria'): as we have seen in the WSU studies, the numbers in experimental groups at the end of the study often do not match those included at the start. Premature death is likely to account for some of the discrepancies but this is rarely made transparent.

The transgenerational studies that are the focus of this paper correspond to cluster randomised trials in medicine: the treatment is allocated to an individual F0 rodent but then all members of a cluster (i.e. a litter) are deemed to be exposed to it. There are extended CONSORT guidelines that deal with cluster randomised trials [24] and these stress the importance

of accounting properly for both numbers of clusters and numbers of individuals in a study over time and, most importantly, accounting for the dependence of observations within a cluster, both in the design (i.e. having a sufficiently powerful study) and in the analysis. Scientists conducting these transgenerational rodent trials could usefully build both these guidelines and the ARRIVE2.0 guidelines into their design and reporting protocols.

One issue dealt with in [23] is the importance of making data available for secondary and re-analysis. For transgenerational studies, these data should include all identifying information for each rodent at each generation so that the kinds of analyses described in this paper can be carried out. This does, of course, require experimenters to keep careful records of all their rodents so that their pedigrees can be reported: "if in doubt, include it" should be a reporting mantra for these studies. All the data used in this paper can be accessed from the links provided in S1–S3 Datasets.

## Conclusions

Rodent litter effects are ubiquitous in transgenerational research and in other multi-generational designs. There is a well-established corpus of statistical models that can be applied in these situations so that power and treatment effects can be properly estimated. Unfortunately, these methods are often not applied and analysis problems are exacerbated in the literature by a lack of transparency about details of the design such as treatment allocation protocols and procedures for dealing with missing data at each stage.

Given the methodological focus of this paper, data from five studies from the same centre [2–6] were amalgamated into a single dataset in order to estimate parameters of interest precisely. It is, however, recognised that more disaggregated analyses might be substantively more appropriate in some circumstances. Analyses of this dataset has filled a substantial gap in the literature by providing estimates of litter effects and showing how they can modify conclusions about treatment effects. Although there are likely to be situations where litter effects are sufficiently small that they do not have an important effect on the conclusions, it is shown here that, for example, over 80% of all the variation in a puberty outcome in fourth generation rats (F3) can be attributed to variation between the mothers at F2 (Table 1). One effect of this is to reduce the sample size of atrazine-exposed rats from an observed value of 114 to an effective size of between 14 and 18, thus severely compromising the power of the study (Table 2). This reduction in power shows up in the estimated effect of atrazine on puberty which is no longer statistically significantly different from zero at conventional levels (Table 3).

Findings such as these are much more than methodological quibbles. There is widespread concern about the effects of herbicides, especially glyphosate, on human health and this concern has spread to US courts [25]. The public needs to be alerted to soundly based health risks from using pesticides of all kinds but they also need to be protected from assertions about risk that are based on questionable evidence [26]. Hopefully, researchers will take on board the issues discussed here so that the design and analysis of rodent studies that examine the transmission of pathologies across generations exhibit a greater degree of methodological rigour. It is, however, worth noting that discussion of litter effects goes back at least 50 years [27] and yet the issue does not appear to have fully penetrated methods courses for life science students. This apparent lacuna in life science methods training is something that might be addressed.

## Supporting information

**S1 Appendix. Definitions of binary outcomes.**
(DOCX)

**S2 Appendix. Specifications of statistical models [28, 29].**
(DOCX)

**S1 Table. Number of F0 females by study, group and generation.**
(DOCX)

**S1 Dataset. WSU1_F1 (Excel; Variable definitions + data).**
(XLS)

**S2 Dataset. WSU1_F2 (Excel; Variable definitions + data).**
(XLS)

**S3 Dataset. WSU1_F3 (Excel; Variable definitions + data).**
(XLS)

## Author Contributions

**Conceptualization:** Ian Plewis.

**Data curation:** Ian Plewis.

**Formal analysis:** Ian Plewis.

**Investigation:** Ian Plewis.

**Methodology:** Ian Plewis.

**Validation:** Ian Plewis.

**Writing – original draft:** Ian Plewis.

**Writing – review & editing:** Ian Plewis.

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
