## [Decision Letter · Decision Letter 0]

25 Jun 2020

PONE-D-20-01881

Pesticides and transgenerational inheritance of pathologies: Designing, analysing and reporting rodent studies.

PLOS ONE

Dear Dr. Plewis,

Thank you for submitting your manuscript to PLOS ONE. After careful consideration, we feel that it has merit but does not fully meet PLOS ONE’s publication criteria as it currently stands. Therefore, we invite you to submit a revised version of the manuscript that addresses the points raised during the review process.

With respect to the manuscript, from myself I would like some of the statements to be less controversial and couched in terms of potential difficulties with the current reporting of animal studies. For example I would change your first sentence in the abstract (pointed out by reviewer 2) to something like " To date studies examining trans-generational transmission of pathologies in rodents has not always taken into account the design and analysis of single centre animal studies", rather than what is written. Given there are many frank comments about the work presented by WSU that could be perceived as being biased, it is in your interests to ensure that nothing is within the manuscript that could be misinterpreted.

We look forward to receiving your revised manuscript.

Kind regards,

Rodney John Scott

Academic Editor

PLOS ONE

Additional Editor Comments:

Your manuscript has been reviewed by two referees and one has more concerns than the other. I have also read your manuscript and find the subject matter interesting and noteworthy. There are, however, some issues that need to be carefully managed especially as one of the reviewers is from Washington State University.

All issues raised by the reviewers must be addressed and as such your response must be indicated in the revised manuscript.

Reviewers' comments:

Reviewer's Responses to Questions

**Comments to the Author**

1. Is the manuscript technically sound, and do the data support the conclusions?

Reviewer #1: Yes

Reviewer #2: Partly

2. Has the statistical analysis been performed appropriately and rigorously? 

Reviewer #1: Yes

Reviewer #2: No

3. Have the authors made all data underlying the findings in their manuscript fully available?

Reviewer #1: Yes

Reviewer #2: No

4. Is the manuscript presented in an intelligible fashion and written in standard English?

Reviewer #1: Yes

Reviewer #2: Yes

5. Review Comments to the Author

Reviewer #1: General comments

Plewis have submitted a paper “Pesticides and transgenerational inheritance of pathologies: Designing, analysing and reporting rodent studies” (PONE-D-20-01881). The paper is describing that many studies on pesticides show weaknesses in design and analysis and a lack of transparency in reporting the results. This paper makes a critical examination of the methodological and statistical aspects of multi-and transgenerational studies in detail and focus is on the estimation of ‘litter effects’. Overall, this study examines published data from a series of widely reported studies carried out at Washington State University that were merged into a single dataset in order to estimate these litter effects and the associated treatment effects. The paper shows that attention should be paid to the design and analysis of multi-generational rodent studies. If these studies of pesticides are not properly designed and evaluated by correct statistical models in human risk assessments, it may have important implications for public health. I find this analysis very relevant for the research area and I have below included comments, alongside other points that I feel needs to be addressed in the revised paper.

Specific comments:

Introduction

p.3

• “The toxicity and carcinogenicity of pesticides are widely assessed via feeding

experiments using rodents”

o Please consider to also include reproductive toxicity and change to “The toxicity including reproductive and carcinogenicity of pesticides are widely assessed via feeding experiments using rodents”

o This as all transgenerational studies are reproductive toxicity studies (while some of them might also look at carcinogenicity)

• a pesticide (two herbicides, two insecticides and one fungicide).

o Please consider to write the names of the compound

p.4:

• Consider describing what Litter effect is before mentioning the different cases. And not only in a parenthesis on page 4 (i.e. the tendency of two randomly selected rodents from the same litter to be more alike than two randomly selected rodents from different litters)

• Describe what happens if this is not taking into account. If the statistical analysis used the individual pups as statistical unit instead of litter this will lead to the use of an excessively high N value in the statistical analysis. This gives an increased risk of false-positive results. Consider referring to:

• R.R. Holson, B. Pearce, Principles and pitfalls in the analysis of prenatal treatment effects in multiparous species, Neurotoxicology and Teratology,Volume 14, Issue 3,1992, Pages 221-228, https://doi.org/10.1016/0892-0362(92)90020-B

Designing transgenerational studies

• When referring to [15] please consider mentioning 1.st author

• Good explanation of the transgenerational study design

Analysing observational dependence

• P.7 “Litter effects arise when the offspring (Fk, k ≥ 1) of pregnant females (F0) are studied”

o Please explain: Fk, k ≥ 1 (what is Fk?) is it the same as Fx?

o Clarify that F1 is the offspring and F0 is the dams/pregnant females

• P.8 “blocking variable” please explain more for non- Statistics readers.

• P.8 Please elaborate and help the reader in understanding ‘xt’ and ‘me’ procedures without reading [18]

a. “All results in this section come from the ‘xt’ and ‘me’ procedures in STATA [18]

• P.9. Please provide more table text for table 1 so it can “stand alone” and help the reader for understanding the different comparisons. Have the comparisons been made for one study only or from controls alone or a mix of treatments?

• P.10: please clarify “less efficient” here.

• P. 11: Table 2. Please abbreviate effective sample size (ESS) in the table text. Please provide more table text for table 2 as well.

• P. 11: please explain why “the litter effects are greater in puberty

• P. 12: please explain the plus and minus signs in the table 3 in the table text

• P. 13. Please explain the text and make a reference “as females are less likely to

be obese according to the method used to determine obesity”.

• p. 13. Please explain “one via the female line and one via the male line”. Is it like a grandmother from maternal and paternal line?

Reporting standards

• p. 15 Agree to the statement “vital that researchers make all their data available”

Conclusion

• P. 15 “for example, that over 80% of all the variation in a puberty outcome in fourth generation rats (F3)” where in the tables are these 80% seen?

• Please consider reflecting on the last sentence “Soundly designed and analysed research might change the position in the future but, for now, the evidence for transgenerational or epigenetic transmission of pathology from pesticide exposure is unconvincing.”

o Is it only due to statistics? Is it only for the studies analysed? Could it be a possibility to make a final statement where it is clear for the reader that improper statistics in transgenerational studies could both give false negatives or positives?

S1 Appendix

Please clarify what the implication of the ‘inconsistencies across the datasets for control rats with the same ID’

S1 Datasets.

Please explain “Lean”

Reviewer #2: 1) The author has expressed a severe bias against epigenetic inheritance and studies involved in the analysis of environmental toxicants such as glyphosate. A scholarly approach and journal such as Plos One requires an unbiased presentation of the data and discussion of the topic and study. An example is the first sentence in the Abstract that made a generalized derogatory statement against the field of epigenetic inheritance and transgenerational studies. With over 3000 publications on the topic and over 100 different laboratories reporting studies in nearly 20 different species, to make a general statement condemning the field demonstrates a bias that is not appropriate in a scholarly publication. Other biased statements that need to be removed include page 2, lines 1-3, 18-20, page 3 lines 17-19, page 4 lines 1-2, 4-5, 21-23, page 5 lines 5-6, 8-9, page 15 lines 12-13, page 16 lines 6-13. In addition, the author in assessing the studies and applying his arguments makes assumptions and subjective statements regarding the studies that are not based on the original studies described information. This indicates experimental design issues such as page 4 lines 2, 15, page 5 lines 1-4, page 6 lines 9, 10, 15-17, page 7 lines 2-5, 8-9, page 15 lines 12-13. The author generally makes overarching comments about the field and transgenerational studies, rather than the studies investigated. The bias against studies investigating pesticide impacts, such as those with glyphosate, are not appropriate and question the potential conflicts of interest of the author and analysis. A thorough revision removing these biases is required.

2) A major deficiency is the author developed an approach to assess litter effects and used data for kidney, testis, ovary or prostate in the supplement S2, but did not report data or reanalyze results of litter effects on any of these pathologies. The assumption is that no litter effects were observed, however, the author needs to more fully disclose all data from the analysis. A negative effect of litter on these pathologies should be reported to put the current study into a better perspective, and fully disclose the results for the study. All data analyses need to be reported, both positive and negative.

3) A major experimental flaw in the analysis is the combination of males and females, as well as the combination of all treatments into one data set used for the analysis. This disregards critical biological issues in male and female responses to specific exposures and differences in disease etiology between the males and females for specific pathologies. The control of puberty and metabolic disease in males and females are often opposite, as well as are the actions of different endocrine disruptors on pathology. Estrogenic compounds promote a premature puberty in females and delayed puberty in males, while androgenic effects are opposite. To combine males and females as an experimental unit negates the biological differences and complicates the disease correlations with litters sought. The combination of some estrogenic exposure and some androgenic exposures and others that are both, also negates the different biological effects of the exposures by combining data. Therefore, the experimental design may generate observations that are distinct from the experiments and randomly derived. To combine these and assess impacts is questionable. What is requested is that the author performs the same analysis on the isolated data sets without the combination of the sex or exposure. This would assess the potential litter effect within an experiment and not bias it by combining sex and exposure. This additional analysis will be useful to determine if combined data versus individual study data without males and females combined, provides distinct observations. The potential litter effects and different approach to the analysis will then provide a more biologically appropriate assessment of the study objective.

4) Many of the exposure studies and pathology were unique to a specific exposure and distinct from other studies. For example, the atrazine exposure promotes a lean phenotype and none of the other exposures do so, while instead many of the others promote an obese phenotype. To combine different exposures and males and females complicates any such analysis. The Tables presented did not provide sufficient information to determine if isolated treatment sets were used, but did appear to combine treatments and males and females. Again, a truly isolated analysis for a specific treatment and exposure is needed to compare the combined data presented now in the manuscript.

5) The puberty and obesity pathologies are not able to be assessed in males and females combined in the analysis as presented in the current study. Since both pathologies are distinct and induced differently by the same exposure, combination of the sexes for analysis is difficult to both interpret and make any biological inferences. New data analysis is needed with the isolated sexes and exposures to compare with the combined data presented.

6) As described for the previous papers’ studies, the F0 generation females were randomly selected for treatments. The author makes the statement that randomization was compromised for these studies because there is an imbalance between the number of controls and treated animals. This seems a marked overstatement, as the Fisher’s Exact test handles this well.

7) The justification for combining the various studies into one analysis seems weak. The Permethrin+DEET study had its own controls and published many years separated from the others. The Methoxychlor study had its own controls and was a few years separated from the others. The Vinclozolin and Atrazine studies shared controls. The Glyphosate study shared some but not all controls with the Vinclozolin and Atrazine studies, and had some of its own. It would improve the manuscript to present the results of analyzing these studies separately, as well as combined.

8) According to the previous papers studies, the F0 generation females were randomly selected and no pathology on the females was performed, so the comments to the contrary by the author should be removed from the manuscript. In addition, the statement on page 7 that an imbalance in treated and control n-value is a failure to randomize is misleading and inaccurate statement. Only three of the studies appear to have shared controls, so this is not a good justification to combine the data for the analysis as suggested by the author. The final statement in the Appendix S1 that common controls were used and information is missing is a presumption by the author that is not appropriate.

9) The Table legends need significantly more information to help interpret the numerical values presented, clarify ranges presented and state males and females combined, as well as treatments. For Table 1, all data appear to be combined for a specific pathology. Due to the male versus female and distinct actions of exposures, misleading data will be obtained when correlations are made for disease. For Table 2, details in the legend are needed on what the different value ranges represent. Tables 3 and 4 legends need to help clarify and define values and ranges presented. Insufficient information is available to have an expert or non-expert interpret what is reported.

10) The Conclusion presented has a bias expressed that is not linked to the data presented or analysis. This conclusion needs to be removed for a more objective and scholarly reflection on the current study analysis and include limitations of the analysis (e.g. combining data sets) that impact the interpretations.

This study presents some good ideas on how to account for ancestral litter relatedness in analyzing data of this type, and how to improve reporting of data including breeding records. If the deficiencies listed above are addressed, this will be a useful addition to the literature.

6. PLOS authors have the option to publish the peer review history of their article (what does this mean?). If published, this will include your full peer review and any attached files.

Reviewer #1: No

Reviewer #2: No

---

## [Author Response · Author response to Decision Letter 0]

21 Jul 2020

See responses to reviewers and cover letter.

---

## [Editor Report · Decision Letter 1]

31 Jul 2020

PONE-D-20-01881R1

Pesticides and transgenerational inheritance of pathologies: Designing, analysing and reporting rodent studies.

PLOS ONE

Dear Dr. Plewis,

Thank you for submitting your manuscript to PLOS ONE. After careful consideration, we feel that it has merit but does not fully meet PLOS ONE’s publication criteria as it currently stands. Therefore, we invite you to submit a revised version of the manuscript that addresses the points raised during the review process.

We look forward to receiving your revised manuscript.

Kind regards,

Rodney John Scott

Academic Editor

PLOS ONE

Additional Editor Comments (if provided):

The concerns of the reviewers have been addressed but there is one statement that I think needs to be stated slightly differently. In the last paragraph of the manuscript there is a sentence that states "The public needs to be alerted to soundly-based health risks from using pesticides of all kinds but they also need to be protected from assertions about risk that are based on flimsy evidence" To ensure there is not an adverse response to this, it would be better if it were softened to: "The public needs to be alerted to soundly-based health risks from using pesticides of all kinds but they also need to be protected from assertions about risk that are based on confounded evidence"

---

## [Editor Report · Decision Letter 2]

16 Sep 2020

Pesticides and transgenerational inheritance of pathologies: Designing, analysing and reporting rodent studies.

PONE-D-20-01881R2

Dear Dr. Plewis,

We’re pleased to inform you that your manuscript has been judged scientifically suitable for publication and will be formally accepted for publication once it meets all outstanding technical requirements.

Kind regards,

Rodney John Scott

Academic Editor

PLOS ONE

Additional Editor Comments (optional):

The concerns raised by the reviewers and editor have been addressed.
---

## [Editor Report · Acceptance letter]

21 Sep 2020

PONE-D-20-01881R2 

Pesticides and transgenerational inheritance of pathologies: Designing, analysing and reporting rodent studies 

Dear Dr. Plewis:

I'm pleased to inform you that your manuscript has been deemed suitable for publication in PLOS ONE. Congratulations! Your manuscript is now with our production department. 

Kind regards, 

on behalf of

Dr. Rodney John Scott 

Academic Editor

PLOS ONE